# A Clinical and Economic Comparison of Cell-Based Versus Recombinant Influenza Vaccines in Adults 18–64 Years in the United States

**DOI:** 10.3390/vaccines12111217

**Published:** 2024-10-26

**Authors:** Myron J. Levin, Neda Al Rawashdh, Liliane Mofor, Pablo Anaya, Richard M. Zur, Emily B. Kahn, Daniel Yu, Joaquin F. Mould-Quevedo

**Affiliations:** 1Departments of Pediatrics and Medicine, Anschutz Medical Campus, University of Colorado, Aurora, CO 80045, USA; 2Real-World Solutions, IQVIA, Falls Church, VA 22042, USA; 3CSL Seqirus USA, Summit, NJ 07901, USA; 4CSL Seqirus Australia, Melbourne 3052, Australia

**Keywords:** cell-based influenza vaccine, recombinant influenza vaccine, cost-effectiveness, United States

## Abstract

Background: This analysis compares the cost-effectiveness of a cell-based influenza vaccine to a recombinant influenza vaccine, and each to no vaccination. The analysis is based on United States (US) commercial and societal perspectives. Methods: A Susceptible–Exposed–Infectious–Recovered (SEIR) transmission model of the total US population followed with a cost-effectiveness model for 18–64-year-olds was used to estimate the clinical and economic impact of vaccination over one influenza season (2018–2019). Deterministic and probabilistic sensitivity analyses were conducted. Results: Both enhanced vaccines prevented a substantial number of influenza cases and influenza-related deaths compared to no vaccination. The cell-based vaccine was associated with higher quality-adjusted life years (QALYs) gained compared to the recombinant vaccine or no vaccination. The cell-based vaccine had a 36% lower vaccination cost, amounting to $2.8 billion in cost savings, compared to the recombinant vaccine. The incremental cost-effectiveness ratios (ICERs) for the cell-based vaccine, compared to the recombinant vaccine or no vaccination, were dominant from all payer perspectives, regardless of risk groups. Conclusions: Overall, the cell-based vaccine was cost-saving compared to the recombinant vaccine for subjects aged 18–64 years in the US, achieving comparable health outcomes with a significant reduction in associated costs.

## 1. Introduction

Influenza is a respiratory viral disease caused mainly by two influenza types: type A (H1N1 and H3N2) and type B (Yamagata and Victoria) [1]. Although most people recover within 2 weeks, some develop life-threatening complications like pneumonia and the worsening of underlying chronic illnesses. People at higher risk of developing serious influenza-related complications include adults aged 65 years and older, people of any age with certain chronic medical conditions (such as asthma and other chronic pulmonary conditions, diabetes, or heart disease), those who are pregnant, children younger than 5 years old, and immune compromised individuals [2].

The World Health Organization estimates a billion cases occur annually worldwide including 2 to 5 million cases of severe illness and 290,000 to 650,000 annual global deaths due to influenza-related respiratory complications alone [1,3]. During the 2022–2023 influenza season in the United States (US), nearly 31 million people were estimated to be sick with influenza, of whom 17 million were adults between 18 and 64 years of age [4]. The Centers for Disease Control and Prevention (CDC) estimates that during the 2022–2023 season, influenza resulted in 14 million visits to a health care provider, 360,000 hospitalizations, and 21,000 deaths [4]. The influenza burden in the US can vary widely as determined by factors such as the characteristics of the circulating strains and vaccine matching, the timing of the season, vaccine effectiveness, and the proportion of the population vaccinated [4]. The annual economic burden of influenza on the US healthcare system has been estimated at $11.2 billion ($6.3–$25.3 billion) [5], with unvaccinated individuals responsible for almost 80% of the financial burden [6].

Vaccination is the cornerstone for prevention and control of influenza. The CDC recommends seasonal influenza vaccination for all individuals aged 6 months and older [7]. A variety of licensed influenza vaccines have been available in the US, from several different manufacturers. These include unadjuvanted egg-based trivalent and quadrivalent inactivated influenza vaccines, adjuvanted trivalent egg-based inactivated influenza vaccines, high-dose trivalent egg-based inactivated influenza vaccines, cell-based vaccines, and recombinant vaccines. However, the effectiveness of these vaccines fluctuate due to various factors [7], such as mutations in the circulating strains, the relative prevalence of type B and type A influenza strains, and immune deficits that prevent some vaccinated individuals from mounting an effective immune response. Egg-based vaccines may contain mutations in viral hemagglutinins that fail to stimulate strong neutralizing antibodies against the circulating strains of influenza [8,9,10].

Non-egg-based vaccine manufacturing methods provide a solution to the issue of acquired mutations during serial passage in eggs. Currently, two non-egg-based vaccines are available in the US: the quadrivalent cell culture-based influenza vaccine (IIV4c) and the recombinant quadrivalent influenza vaccine [7]. These vaccines may provide additional protection compared to traditional egg-based vaccines; they are more likely to represent the viral strains targeted by the seasonal vaccine [11,12]. During the 2023–2024 season, all vaccines administered in the US were quadrivalent [13]. However, due to the absence of the Yamagata/B subtype worldwide since March 2020, the US Food and Drug Administration advisory committee recommended only trivalent influenza vaccines for the upcoming season [14].

The overall effectiveness of the seasonal vaccine for the 2022–2023 season was 46%. However, this effectiveness varied by age group, being highest for children 6 months to 4 years (54%) and for adults aged 18–65 years (48.3%). Vaccination coverage rates ranged from 44% to 70% depending on the age group. Influenza vaccination was estimated to prevent 6.0 million influenza-related illnesses, 2.9 million medical visits, 65,000 hospitalizations, and 3700 deaths [4].

Real-world studies during the 2019–2020 season showed that the cell-based vaccine was significantly more effective than the egg-based vaccine in preventing influenza-related hospitalization visits (relative vaccine effectiveness [rVE] = 9.3%, 95% CI: 0.4% to 17.3%), ER visits (rVE = 6.8%, 95% CI: 2.1% to 11.3%), and outpatient visits (rVE = 10%, 95%, CI: 2.7 to 16.7%) [15,16,17]. However, no statistically significant differences were found between the recombinant vaccine and egg-based standard dose vaccines in preventing hospitalization among adults 18–64 years (rVE was 9.1% (95% CI: −20.8% to 31.6%) [18]) and hospital-based clinic or emergency department visits (rVE was 9.0% (95% CI: −30% to 35%) [19].

Understanding the cost-effectiveness profile of non-egg-based influenza vaccines is essential to guide payers in allocating resources efficiently and prioritizing vaccines that offer the best health outcomes for the funds available. Due to the lack of published evidence, we conducted this study to evaluate the cost-effectiveness of the cell-based influenza vaccine compared to the recombinant influenza vaccine—focusing on the prevention of influenza-related outpatients visits, emergency department (ED) visits, and hospitalizations among US adults aged 18–64 years—from the US commercial payer and societal perspectives using vaccine effectiveness for the 2018–2019 season.

## 2. Methods

### 2.1. Model Design

#### 2.1.1. Transmission Model Structure

A Susceptible–Exposed–Infectious–Recovered (SEIR) compartmental model was developed using Microsoft^®^ Excel^®^ for Microsoft 365. This model, which simulates influenza transmission among vaccinated and unvaccinated individuals at varying levels of risk for influenza complications, was used to estimate the number of infected individuals aged 18 to 64 years within the US population during the 2018–2019 season. Since effectiveness data for cell-based and recombinant vaccines were available for both the 2018–2019 and 2019–2020 seasons, we used the 2018–2019 season as the base case and conducted a scenario analysis for the 2019–2020 season.

In the SEIR model:•“S” represents susceptible individuals who are at risk of contracting influenza.•“E” represents individuals who have been infected but are not yet infectious (i.e., in the latent phase).•“I” represents infectious individuals who can transmit influenza.•“R” represents individuals who have recovered and thus gained immunity during that influenza season or died from influenza.

In the model, all individuals start in the susceptible population. Those who are vaccinated become immune and are removed from the susceptible state. If the vaccine is not fully effective, some vaccinated individuals who are exposed to the virus will transition to the next (infectious) state. In our analysis, we stratified the population into three age groups: children (0 to 17 years old), adults (18 to 64 years old), and older adults (65 years and older). The age group-specific rate of movement from susceptible to infected was calculated based on the reproductive number (R0), the contact matrix structure, and the estimated daily infectivity rates of influenza A and B viruses [20,21,22].

#### 2.1.2. Cost-Effectiveness Model Structure

A decision tree cost-effectiveness model (CEM) was built that incorporated probabilities and costs associated with disease outcomes (Figure 1). This model used the number of adults aged 18 to 64 years from each SEIR compartment to calculate the overall number of symptomatic and asymptomatic individuals (children, adults, and elderly) infected with influenza during one influenza season (i.e., one year). The resource use and cost for symptomatic adult patients were estimated based on the probabilities of having an outpatient visit, an ED visit, being hospitalized, or dying from influenza. Cases were categorized based on the risk (high or low) of developing influenza-related complications (Figure 1). High risk was identified by the presence of one or more chronic conditions, including cardiovascular, respiratory, renal, or immune-related diseases. Low risk was identified if patients did not have any of the high-risk chronic conditions [19]. The CEM compared influenza-related outcomes for US adults who were unvaccinated, vaccinated with a cell-based vaccine, or vaccinated with a recombinant vaccine. The numbers of outpatient visits, ED visits, hospitalizations, and deaths, as well as life years and quality-adjusted life years (QALYs) for each vaccine type were calculated based on vaccine effectiveness estimates. Direct medical costs and vaccination costs from the commercial perspective, as well as indirect costs associated with productivity losses from the societal perspective, were calculated for each vaccine type. Incremental cost-effectiveness ratios (ICERs) were calculated between the cell-based vaccine and recombinant vaccine, and between each of these vaccines and no vaccine.

### 2.2. Model Inputs

#### 2.2.1. Population and Epidemiology Inputs

The age distribution of the US population was obtained from the 2024 US Census data [23]. The contact matrix, representing interactions between different age groups, was derived from a contact matrix for Germany, which we assumed, based on expert opinion, most closely resembled the contact patterns of the US population (Appendix A) [20]. The probability of transmission from an infectious contact was estimated using a transmission rate (R0) of 1.28 based on a review of the literature and commonly used estimates of R0 for seasonal influenza, [21] and the daily infectivity rate of influenza viruses (A and B) during the infection period [22] (Table 1). It was assumed that all individuals with influenza, including asymptomatic ones, were equally infectious [24]. Infected patients were presumed to recover after an average of 7 days (ranging from 3 to 14 days) of infection, subsequently gaining immunity if they were not vaccinated [25].

**Table 1 vaccines-12-01217-t001:** Population and clinical inputs.

Variable	Base Case Analysis Value	Assumed Range * or 95% CI Used in Sensitivity Analyses	Reference
Population and epidemiological inputs			
Total US population	341,963,408	NA	US Census Bureau 2024 [23]
Population aged 0–17 years	74,889,986	NA	US Census Bureau 2024 [23]
Population aged 18–64 years	205,178,045	NA	US Census Bureau 2024 [23]
Population aged 65 years or more	61,895,377	NA	US Census Bureau 2024 [23]
Percentage of population at low risk of complications	80.5%	64%, 97% *	Zimmerman 2010 [26]
Percentage of population at high risk of complications	19.5%	16%, 23% *	Zimmerman 2010 [26]
New infections per case (R0)	1.28	1.19, 1.37 *	White 2014 [21]
Infectivity rate per day		NA	Tsang 2015 [22]
Day 1	0.0%		
Day 2	0.5%		
Day 3	24.1%		
Day 4	35.0%		
Day 5	29.0%		
Day 6	9.5%		
Day 7	2.1%		
Clinical/disease burden inputs			
Percentage of symptomatic adult influenza cases with outpatient visits			
Low risk	39.8%	31.8%, 47.8% *	CDC 2024 [4]
High risk	47.3%	37.8%, 56.8% *	DeLuca 2023 [27]
Percentage of symptomatic adult influenza cases with ED visits			
Low risk	2.3%	1.8%, 2.8% *	Agency for Healthcare Research and Quality (AHRQ) 2024 [28]
High risk	4.8%	3.8%, 5.8% *	Agency for Healthcare Research and Quality (AHRQ) 2024 [28]
Percentage of symptomatic adult influenza cases with influenza-related hospitalization			
Low risk	0.8%	0.6%, 1.0% *	Dolk 2016 [29], CDC 2024 [4]
High risk	3.2%	2.6%, 3.9% *	DeLuca 2023 [27]
Percentage of hospitalized patients requiring ICU care			
Low risk	16.2%	12.9%, 19.4% *	Sumner 2023 [30]
High risk	28.0%	22.4%, 33.6% *	Macias 2021 [31]
Percentage of ICU patients requiring mechanical ventilation	38.8%	31.0%, 46.6% *	Sumner 2023 [30]
Mean duration of hospitalization (days)			
Low risk (no ICU)	3.4	2.7, 4.1 *	HCUP 2023 [32]
High risk (no ICU)	5.3	4.2, 6.4 *	HCUP 2023 [32]
ICU patients—no MV	7.9	6.3, 9.4 *	Dasta 2005 [33]
ICU patients with MV	14.4	11.5, 17.3 *	Dasta 2005 [33]
Probability of long-term sequelae (low-risk population)	0.70%	0.56%, 0.84% *	DeLuca 2023 [27]
Probability of long-term sequelae (high-risk population)	3.55%	2.84%, 4.26% *	Macias 2021 [31]
Probability of death in low-risk population	0.04%	0.03%, 0.05% *	Dolk 2016 [29]
Probability of death in high-risk population	0.73%	0.58%, 0.88% *	Dolk 2016 [29], CDC 2024 [4]
Vaccination rate and vaccine effectiveness			
Vaccination coverage in US			CDC 2024 [34]
Vaccination coverage in children 0–17 years old	57.4%	45.9%, 68.9% *	
Vaccination coverage in adults 18–64 years old	40.0%	32.0%, 48.0% *	
Vaccination coverage in adults ≥ 65 years old	69.7%	55.8%, 83.6% *	
Rate of vaccination per month for vaccinated population			Divino 2021 [15]
August	1.3%	NA	
September	17.8%		
October	45.9%		
November	19.0%		
December	10.2%		
January	5.8%		
Average vaccine effectiveness in preventing infections			Nolan 2021 [35]
Effectiveness in children (age 0–17 years)	54.6%	43.7%, 65.5% *	
Effectiveness in adults (age 18–64 years)	70.3%	56.2%, 84.4% *	Brogan 2017 [36]
Effectiveness in adults (age 65+ years)	58.0%	46.4%, 69.6% *	Brogan 2017 [36]
Relative effectiveness of non-egg vaccines vs. standard vaccine for 2018–2019 season			
Preventing outpatient visits			
Cell-based vaccine	12.5%	95% CI: 4.7%, 19.6%	Stein 2024 [16]
Recombinant vaccine	7.0%	95% CI: −60%, 46%	Zimmerman 2023 [19]
Preventing ED visits			
Cell-based vaccine	6.5%	95% CI: 0.1%, 12.5%	Krishnarajah 2021 [37]
Recombinant vaccine	7.0%	95% CI: −60%, 46%	Zimmerman 2023 [19]
Preventing hospitalizations			
Cell-based vaccine	6.5%	95% CI: 0.1%, 12.5%	Krishnarajah 2021 [37]
Recombinant vaccine	7.0%	95% CI: −26.8%, 31.8%	Hsiao 2023 [18]
Relative effectiveness of non-egg vaccines vs. standard vaccine for 2019–2020 season			
Preventing outpatient visits			
Cell-based vaccine	10%	95% CI: 2.7%, 16.7%	Stein 2024 [16]
Recombinant vaccine	9.0%	95% CI: −30%, 35%	Zimmerman 2023 [19]
Preventing ED visits			
Cell-based vaccine	6.8%	95% CI: 2.1%, 11.3%	Divino 2021 [15]
Recombinant vaccine	9.0%	95% CI: −30%, 35%	Zimmerman 2023 [19]
Preventing hospitalizations			
Cell-based vaccine	9.3%	95% CI: 0.4%, 17.3%	Imran 2023 [17]
Recombinant vaccine	9.1%	95% CI: −20.8%, 31.6%	Hsiao 2023 [18]

**Abbreviations**: CDC, Centers for Disease Control and Prevention; ED, emergency department; HCUP, Healthcare Cost and Utilization Project; ICU, intensive care unit; MV, mechanical ventilation; NA, not available. **Notes**: * Assumed Range. (1) Long-term sequelae include cardiac events (e.g., acute myocardial infarction) and respiratory events (e.g., acute respiratory event or exacerbation of COPD) or neurological events. (2) rVE of recombinant and cell-based vaccines were calculated using inverse probability weight methods in the literature sources. (3) rVE for the recombinant vaccine (Zimmerman 2023 [19]) for outpatient visits and ED visits were reported as a single number and thus was used for both outcomes in the model. (4) Range was calculated by varying point estimate by 20%.

**Table 2 vaccines-12-01217-t002:** Costs and utilities inputs.

Variable	Base Case Analysis Value	Assumed Range * or 95% CI Used in Sensitivity Analyses	Reference
Direct and indirect cost inputs			
Outpatient visit cost	$107.10 (calculated as CMS OP visit cost of $83.02 × commercial insurer payment of 129% of Medicare OP physician fee service)	$85.70, $128.50 *	Congressional Budget Office 2022 [38], CMS physician fee schedule CPT 99203 [39]
ED visit cost	$1870.00	$1496, $2244 *	KFF 2022
Hospitalization cost per day			
Low risk	$3746.18	$2996.94, $4495.42 *	HCUP 2023 [32]
High risk	$4234.40	$3387.52, $5081.28 *	HCUP 2023 [32]
ICU stay with MV	$12,636.00	$10,108.80, $15,163.20 *	Dasta 2005 [33]
ICU stay without MV	$12,398.40	$9918.72, $14,878.04 *	Dasta 2005 [33]
Average ten-year cost of long-term sequelae following influenza-related hospitalization in patients without comorbidities	$36,857.73	$29,486.18, $44,229.27 *	Larsen 2022 [40]
Average ten-year cost of long-term sequelae following influenza-related hospitalization in patients with comorbidities	$72,457.81	$57,966.25, $86,949.37 *	Larsen 2022 [40]
Flucelvax (WAC per dose 2023–2024)	$30.10	$24.08, $36.12 *	RedBook Micromedex 2023 [41]
Flublok (WAC per dose 2023–2024)	$64.51	$51.61, $77.41 *	RedBook Micromedex 2023 [41]
Vaccine administration cost	$31.14	$24.91, $37.37 *	CMS2024 [42]
Indirect cost of reduced productivity at work—per day (USD 2023)	$223.60	$178.88, $268.32 *	US Bureau of Labor Statistics 2024 [43]
Annual loss of earnings due to long-term sequelae	$3018.00	$2414.40, $3621.60 *	Rojanasarot 2023 [44]
Loss of earnings due to influenza-related death	$2,184,687.00	$1,747,749.60, $2,621,624.40 *	DeLuca 2023 [27]
Missed workdays due to illness			
Patient with mild influenza: non-medically-attended	0.5	0.4, 0.6 *	DeLuca 2023 [27], Zumofen 2022 [45]
Patient with outpatient or ED visit	2.5	2.0, 3.0 *	DeLuca 2023 [27], Zumofen 2022 [45]
Low-risk patient hospitalized in general ward	3.4	2.7, 4.1 *	DeLuca 2023 [27], Zumofen 2022 [45]
High-risk patient hospitalized in general ward	5.3	4.2, 6.4 *	DeLuca 2023 [27], Zumofen 2022 [45]
ICU patients without MV	7.9	6.3, 9.4 *	Dasta 2005 [33]
ICU patients with MV	14.4	11.5, 17.3 *	Dasta 2005 [33]
Utility inputs			
Baseline utility value for low-risk individual	0.945		Meier 2015 [46]
Baseline utility value for high-risk individual	0.865	0.69, 1.00 *	Meier 2015 [46]
Symptomatic influenza infection, not medically-attended	−0.23 (for 7 days)	−0.18, −0.28 *	Dolk 2016 [29]
Patients with outpatient or ED visit	−0.455 (for 7 days)	−0.36, −0.55 *	Dolk 2016 [29]
Patient hospitalized due to influenza	−0.628 (for 21 days)	−0.50, −0.75 *	Dolk 2016 [29]
Utility of long-term sequelae (cardio or respiratory events)	0.70	0.56, 0.84	Marcoff 2009 [47], Hamel 2000 [48]

**Abbreviations**: ED, emergency department; HCUP, Healthcare Cost and Utilization Project; ICU, intensive care unit; MV, mechanical ventilation; NA, not available; WAC, wholesale acquisition cost. **Notes**: *Assumed range. (1) Long-term sequelae include cardiac events (e.g., acute myocardial infarction) and respiratory events (e.g., acute respiratory event or exacerbation of COPD) or neurological events. (2) Range was calculated by varying point estimate by 20%.

#### 2.2.2. Clinical/Disease Burden Inputs

The model assumed that patients with asymptomatic influenza did not seek medical attention or develop influenza-related complications. Those with symptomatic influenza could have opted for self-treatment, sought treatment at an outpatient clinic, visited the ED without subsequent hospital admission, or been hospitalized. Hospitalized patients could have been admitted to either the general department or the Intensive Care Unit (ICU), depending on the severity of their symptoms or complications. Patients in the ICU might have required mechanical ventilation if they developed severe lower respiratory complications. Inpatients were at risk of developing long-term sequelae, with the probabilities derived from the literature based on their risk group (low risk and high risk) [27,31]. The incidence rates of outpatient visits, ED visits, hospitalizations, ICU admissions, mechanical ventilation, and influenza-related deaths were obtained from various published sources [4,27,28,29,30,31]. These sources reported the proportion of patients with symptomatic illness by risk status (high risk vs. low risk), using data from the influenza seasons spanning 2016 to 2023 (Table 1).

#### 2.2.3. Vaccination Rate and Effectiveness Inputs

US vaccination coverage rate data for the 2022–2023 season from the CDC [29] were used to estimate the number of vaccinated individuals in each age group, either vaccinated with a cell-based or recombinant vaccine. We calculated the monthly proportion of people vaccinated (from August through to January) using monthly vaccination data from the 2019–2020 season [15] to derive an estimated daily number of vaccinated individuals. Vaccine-specific data on the effectiveness against infection were lacking, therefore we assumed the same VE against infection for both the cell-based and recombinant vaccines, using data from a randomized clinical trial of the cell-based vaccine for the 0–17 years age group, and from a real-world study of egg-based quadrivalent vaccines for adults aged 18–64 years and 65 years and older (Table 1) [18,35,36]. Vaccinated individuals, for whom the vaccine was effective, gained immunity against influenza for one season, with full immunity being attained 14 days post-vaccination [49].

The model used estimates extracted from real-world studies for the cell-based and recombinant vaccines compared to the standard egg-based vaccines [18,19,37,50,51] of relative vaccine effectiveness (rVE) against medically-attended influenza outcomes (influenza-related outpatient visits, ED visits, and hospitalizations) during the 2018–2019 (base case analysis) and 2019–2020 (sensitivity analysis) influenza seasons. Absolute vaccine effectiveness for each was calculated using the following equation, separately for cell-based and recombinant vaccines:*Absolute vaccine effectiveness* = *egg-based vaccine effectiveness* + (*rVE* × *egg-based vaccine effectiveness*).

Using this equation, the absolute vaccine effectiveness is calculated separately for the cell-based and recombinant vaccines, and the rVE compares the efficacy of the cell-based or recombinant vaccines to the egg-based vaccine.

#### 2.2.4. Cost Inputs

Direct medical costs included those associated with influenza-related outpatient visits, ED visits, hospitalizations, over-the-counter medications, and the management of long-term sequelae following influenza-related hospitalization in patients with or without comorbidities. Costs were extracted from a mix of government resources, a drug cost database, and the scientific literature. The detailed list of costs, their ranges for sensitivity analyses, and sources, are included in Table 1. Indirect costs associated with lost or reduced productivity were calculated based on missed workdays due to illness for influenza-related deaths, patients with mild influenza (non-medically attended), patients who had outpatient or ED visits, and patients hospitalized in a general ward or in the ICU (Table 2). When applicable, costs were inflated to 2024 USD using the Medical Care inflation data from the consumer price index [52].

#### 2.2.5. Utilities Inputs

Baseline health utilities were extracted from the published literature [29,46,47,48] for both healthy individuals at low risk and individuals at high risk of developing complications (Table 2). Utility decrements due to medically attended illnesses (outpatient, ED, and hospitalizations) and non-medically attended illnesses were also considered. Additionally, the utility for patients who developed long-term sequelae after influenza was considered.

### 2.3. Scenario Analyses

Several scenario and subgroup analyses investigated the cost-effectiveness of cell-based vaccines compared to recombinant vaccines. In the first scenario, we followed patients for a period of ten years to examine the effects of long-term sequelae following an influenza infection on both health and economic outcomes. The second scenario used the 2019–2020 season vaccine effectiveness estimates to assess the impact of seasonal variability in the influenza virus. In the third scenario, we separately evaluated the cost-effectiveness for adults classified as low and high risk to develop influenza-related complications.

### 2.4. Sensitivity Analyses

One-way sensitivity analyses (OWSA) were carried out to measure the impact of uncertainty on the model results. To explore the impact of variability of the input value, we varied each input variable used in the base case analysis by ±20% or by the 95% confidence intervals found in the published literature. We plotted the ten ICER results with the broadest ranges in a tornado diagram to identify the most influential variables. We conducted probabilistic sensitivity analyses (PSAs) using a Monte Carlo simulation with 1000 iterations, considering the variability around the parameter inputs in the model. We varied the transition probabilities, costs, and utilities using log normal, gamma, and beta distributions, respectively. PSA results are reported in the quadrants of the cost-effectiveness scatter plot.

## 3. Results

### 3.1. Base Case Analyses (2018–2019 Season)

The base case scenario in the full cohort of US adults aged 18–64 years demonstrated that compared to no vaccination, both cell-based and recombinant vaccines reduced the number of influenza cases, outpatient visits, ED admissions, hospitalizations, and deaths. Compared to no vaccination and with an assumption of the same vaccine effectiveness, both the cell-based and recombinant vaccines averted 43,859,447 influenza cases, which equates to preventing 213 cases per 1000 US adults. The cell-based vaccination, when compared to no vaccination, prevented 26,993,917 outpatient visits, 1,758,548 ED visits, and 707,214 hospitalizations. On the other hand, the recombinant vaccine averted 26,558,981 outpatient visits, 1,761,064 ED visits, and 707,920 hospitalizations compared to no vaccination. Further, the cell-based vaccine prevented an additional 434,936 outpatient visits compared to the recombinant vaccine. Both vaccines were able to prevent 15,780 additional deaths compared to no vaccination. The base case analyses showed that the cell-based vaccine reduces the impact of influenza compared to the recombinant vaccine or no vaccination (Appendix A).

For commercial payers, the vaccination of adults aged 18–64 years with the cell-based vaccine saved $14.24 billion compared to no vaccination, whereas the same population, when vaccinated with the recombinant vaccine, saved $11.38 billion (Table 3). The savings were mainly due to differences in cost between vaccines followed by reductions in outpatient costs for patients receiving the cell-based vaccine ($2.89 billion) and the recombinant vaccine ($2.84 billion), with the cell-based vaccine saving $0.05 billion more compared to the recombinant vaccine (Table 3). Between the two vaccines, the cell-based vaccine saved $2.85 billion more than the recombinant vaccine. This was mainly driven by the lower vaccination costs of the cell-based vaccine, which saved $2.82 billion. Furthermore, a reduction in outpatient visits led to an additional saving of $0.05 billion (Table 3).

From a societal perspective, using cell-based and recombinant vaccines resulted in lower direct and indirect costs compared to no vaccination. The direct costs included the expenses related to the vaccination and influenza treatment, whereas the indirect costs were associated with productivity loss due to death and work absence. Compared to no vaccination, the cell-based vaccine led to a savings of $207.28 billion, approximately $2.49 billion more than the recombinant vaccine savings of $204.79 billion (Table 3). Compared to the no vaccination scenario, the cell-based vaccine scenario gained 1,244,883 QALYs and the recombinant vaccine scenario gained 1,243,068 more QALYs, a difference of 1815 QALYs (Table 3). As vaccination was associated with greater cost savings and gains in QALYs compared to no vaccination, the ICERs for both vaccines were dominant from both the commercial payer and societal perspectives (Table 3). The ICER for the cell-based vaccine versus the recombinant vaccine was dominant from both the commercial payer and societal perspectives.

### 3.2. Sensitivity Analyses Results

The PSA provided the distribution of the incremental vaccine costs and incremental effectiveness (QALYs) and are depicted as a scatter plot in Figure 2. From a commercial perspective, for the cell-based vaccine versus no vaccine (Figure 2A), the 1000 PSA simulations were split between the southeast quadrant (82%), denoting a more effective and less expensive intervention, followed by 18% in the northeast quadrant, denoting a more effective and more expensive intervention. Likewise, the PSA simulations for the cell-based vaccine versus recombinant vaccine were distributed mainly in the southeast quadrant (56%) and the southwest quadrant (40%).

Tornado diagrams based on the OWSA showed the sensitivity of the ICER for the cell-based vaccine compared to no vaccine (Appendix A), and the cell-based vaccine compared to the recombinant vaccine (Appendix A). The OWSA indicated that cell-based vaccine remained dominant with ±20% variation in all variables, with the exception of basic reproduction number. With a 20% reduction in R0 (1.02), the cell-based vaccine was no longer dominant, but remained cost-effective compared to the recombinant vaccine at a willingness-to-pay threshold of $50,000 with an ICER of $16,888. In addition to R0, other epidemiologic parameters, including risk of hospitalization and proportion of influenza cases admitted to hospital care were identified as the most influential parameters.

### 3.3. Scenario Analysis

The first scenario analysis explored the vaccine impact over a longer time horizon. Over ten years, the cell-based vaccine scenario reduced the number of patients who developed long-term sequelae by 65% compared to no vaccination. Consequently, the total costs were lower with the cell-based vaccine than for no vaccination or with the recombinant vaccine (Appendix A). The cell-based vaccine was associated with a gain of 1.9 million QALYs more than no vaccination, and 1789 more QALYs gained compared to the recombinant vaccine.

The second scenario analysis used the 2019–2020 influenza season’s vaccine effectiveness of 45% against medically attended, laboratory-confirmed influenza cases (2% lower than the estimate of 47% for the 2018–2019 season.) The relative vaccine effectiveness for the cell-based vaccine compared to the recombinant vaccine for the 2019–2020 season resulted in large differences in the costs saved ($2.82 billion) and QALYs gained (314). The ICERs of the cell-based vaccine were dominant over the recombinant vaccine or no vaccination (Appendix A). The cell-based vaccine was associated with fewer influenza cases, outpatient visits and ED visits, and hospital admissions for both low-risk and high-risk adults when compared to no vaccine (Appendix A).

The third scenario considered the vaccine impact stratified by the risk of developing influenza-related complications (Appendix A). Scenario analysis 3a, for low-risk individuals, found a total cost savings of $5.85 billion for the cell-based vaccines, which was approximately $2.30 billion more than the recombinant vaccines (which saved $3.55 billion). Cell-based vaccines saved 1589 more QALYs than recombinant vaccines, with either vaccine saving over 930,000 QALYs compared to no vaccination. The ICERs for cell-based vaccines were dominant compared with the recombinant vaccine and no vaccination for low-risk adults. For high-risk adults, the cost savings of cell-based vaccination was $8.38 billion and $0.55 billion compared to no vaccination and recombinant vaccines, respectively. The cell-based vaccine saved 226 more QALYs than the recombinant vaccine, and either vaccine saved over 309,000 QALYs compared to no vaccination. Therefore, the ICERs for cell-based vaccines were dominant compared with the recombinant vaccine and no vaccination for high-risk adults.

## 4. Discussion

Non-egg-based influenza vaccines have been approved for vaccinating children, adults, and the elderly. Several real-world studies have shown improved clinical outcomes with these vaccines compared to egg-based vaccines. However, no evidence has compared the effectiveness and economic consequences between the two non-egg-based vaccines (cell-based vs. recombinant vaccine). Therefore, we aimed to conduct cost-effectiveness analyses from the US payer and societal perspectives in adults aged 18 to 64, with or without high risk of developing complications if infected, using vaccine effectiveness data from the 2018–2019 season as the base case.

In this study, an SEIR dynamic transmission model was combined with a decision tree CEM to estimate the health and economic outcomes of no vaccination or vaccination with either recombinant or cell-based vaccines. The cost and effectiveness impact of the vaccination strategy was calculated for the 2018–2019 season, specifically for the adult US population, aged 18 to 64 years, and separately for adults at low or high risk of developing influenza-related complications. There are two main findings of this study. First, compared to no vaccination, vaccinating the US population (assuming average coverage levels ranging between 40% and 70% in all age groups) resulted in preventing 43.85 million influenza cases and approximately 15,780 deaths in adults. Cell-based vaccination reduced the burden of outpatient visits, ED visits, and hospitalizations by approximately 78%, 76%, and 67%, respectively, compared to no vaccination. In comparison with recombinant vaccination, cell-based vaccination led to 399,337 fewer outpatient visits. Second, this study found that cell-based vaccination resulted in a reduced burden of influenza cases, leading to lower direct overall costs for US commercial payers and higher benefits in terms of QALYs gained compared to no vaccination (−$14.24 billion, 1.245 million QALYs) or vaccination with the recombinant vaccine (−$2.85 billion, 1815 QALYs). This resulted in the cell-based vaccine having ICERs that are dominant to the comparators.

A recent cost-effectiveness study compared the recombinant vaccine to a standard dose egg-based vaccine using similar methods to our study; however, our study included a dynamic transmission model that accounts for different levels of contact by age group and for the indirect effects of herd immunity [53]. Nowalk et al. also analyzed data adjusting for the risk of complications rather than stratifying by such risk. Using a similar cost-effectiveness model structure, data inputs, and assumptions, the authors found that compared to the standard dose vaccine, the recombinant vaccine was cost effective; however, cost-effectiveness was likely to be better for older patients and those at higher risk [53].

No published cost-effectiveness studies directly compared cell-based and recombinant vaccines in the US adult population. However, a comprehensive review of 11 cost-effectiveness analyses from nine different countries reported greater cost savings for cell-based vaccines than conventional vaccines [54]. This conclusion by Fisman et al., remained unaffected by the study design, inputs, perspectives, and geographical regions, including both wealthy and low-to-middle income countries. Among specific studies assessing cell-based vaccines, a retrospective cohort study focusing on individuals aged 4 to 64 years found that the cell-based vaccine was associated with significantly lower annual all-cause total costs per patient compared to the egg-derived quadrivalent influenza vaccines ($6769 vs. $7236, *p* < 0.0001). This cost difference of $467 was primarily driven by significantly lower outpatient and inpatient medical costs [15]. Two cost-effectiveness studies from Spain evaluated the recombinant vaccine versus the quadrivalent egg-based vaccine in adults. While one study reported that the recombinant vaccine was associated with lower mortality, hospitalizations, and clinic visits, resulting in cost savings [55], the other study found no cost-effectiveness for recombinant vaccines in adults aged 65 years and older [56]. A study by Nguyen et al. (2021) [57] found the cell-based vaccine to be cost-effective compared to the conventional vaccine among US adults aged 18 to 64 years with significantly reduced outpatient visits, hospitalizations, and deaths. This led to substantial savings in healthcare costs. A study by Deluca et al. (2023) assessed the cost-effectiveness of conventional vaccination versus no vaccination among US adults at a higher risk of influenza complications [27]. The analysis from Deluca et al. found cost savings for vaccination of high-risk individuals compared to no vaccination, which matches the results of our scenario analysis.

This analysis underscores the potential public health benefits of vaccinating adults aged 18–64 years with a cell-based influenza vaccine in the US. The benefit was evident in one influenza season as well as in a longer 10-year follow-up. The estimates indicate that vaccinating all adults, regardless of their risk of developing influenza-related complications, with the cell-based vaccine significantly reduces costs for third-party payers and society. This results in higher clinical benefits and dominant ICERs for the cell-based vaccine. The robustness of the model and its estimates is demonstrated by the fact that the conclusions remained unchanged when all the model parameters (except R0) used in the base case analysis were varied by ±20% in the sensitivity analyses. But even with the variation of R0, the cell-based vaccine was a cost-effective alternative with a willingness to pay below $50,000. A recent update by the European Center for Disease Control on the systematic review and meta-analysis of seasonal influenza vaccines also confirms our analysis assumption that for patients aged between 18–49 years and 50–64 years, the relative effectiveness of the recombinant vaccine was not statistically significant in reducing confirmed influenza cases or influenza-related hospitalizations compared to the standard vaccine [58].

Our analysis has certain limitations. Due to the lack of head-to-head data for cell-based and recombinant vaccines, we assumed that both have the same level of effectiveness against infections. We differentiated their effectiveness in preventing medically attended cases and hospitalizations. Vaccine effectiveness estimates were combined for clinic visits and ED visits and reported for the recombinant vaccine, while estimates for hospitalizations and ED visits were combined and reported for the cell-based vaccine. Therefore, we assigned the same vaccine effectiveness to each outcome based on the available evidence. This study did not differentiate between the symptoms or duration of illness in vaccinated and unvaccinated cases due to limited data availability. However, if vaccines reduce the duration of illness, the benefits of vaccination are expected to be greater.

The overall findings from these cost-effectiveness analyses show that the cell-based vaccine is associated with higher gains in QALYs compared to the recombinant vaccine or no vaccination, which was driven by a reduction in outpatient visits and ED visits. The cell-based vaccine is associated with a 36% greater reduction in vaccination costs compared to the recombinant vaccine, amounting to $2.8 billion in savings on vaccination and administration costs. From the US commercial payer and societal perspectives, the cell-based vaccine, compared to no vaccination, is associated with lower total costs regardless of the risk of influenza complications. This is driven by a 66% reduction in hospitalization costs per season, amounting to $13.1 billion in cost savings. Consequently, the ICERs for the cell-based vaccine, compared to the recombinant vaccine or no vaccination, are dominant from all payer perspectives, regardless of risk groups. Overall, the cell-based vaccine was cost-saving compared to the recombinant vaccine for subjects aged 18–64 years in the US, achieving comparable health outcomes with a significant reduction in associated costs.

## Figures and Tables

**Figure 1 vaccines-12-01217-f001:**
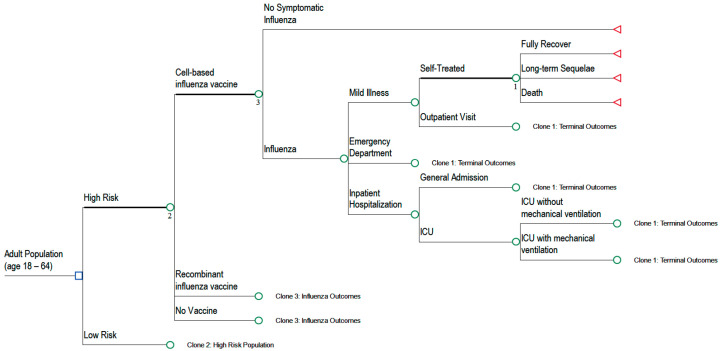
Cost-effectiveness decision tree structure.

**Figure 2 vaccines-12-01217-f002:**
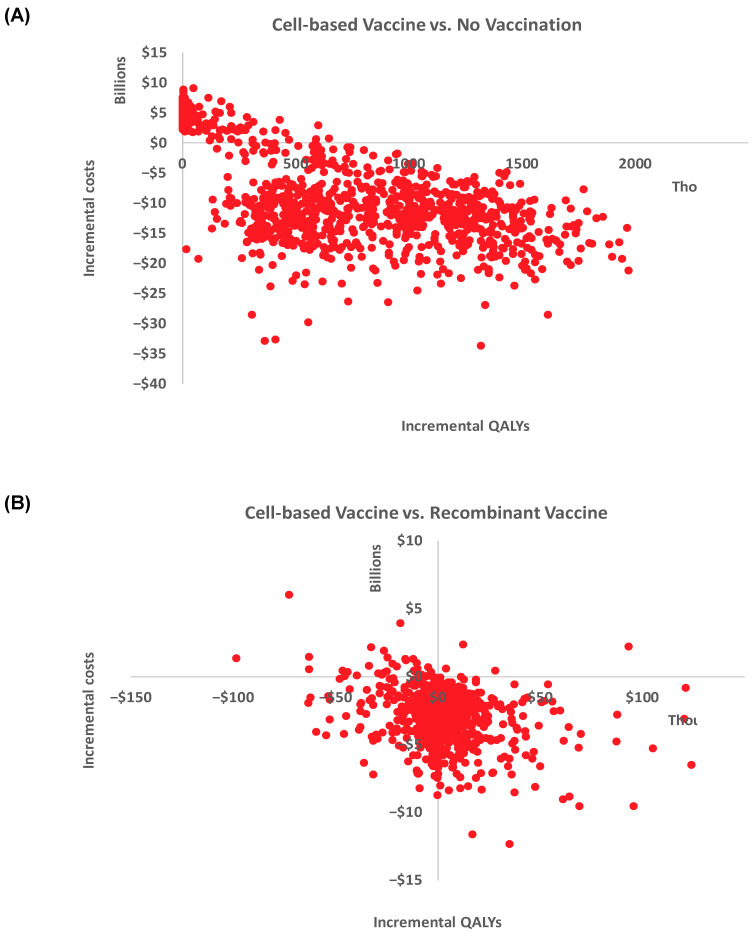
Cost-effectiveness planes for the 2018–2019 season depicting the incremental cost-effectiveness of the cell-based vaccine compared to no vaccine (**A**) and the recombinant vaccine (**B**) from the US commercial perspective over 1000 iterations.

**Table 3 vaccines-12-01217-t003:** Cost-effectiveness results for the 2018–2019 season.

	Cell-Based vs. No Vaccine	Recombinant vs. No Vaccine	Cell-Based vs. Recombinant Vaccine
US commercial payer perspective			
Cost of vaccines and administration	$5.03 B	$7.85 B	−$2.82 B
OP visits costs	−$2.89 B	−$2.84 B	−$0.05 B
ED visits costs	−$3.29 B	−$3.29 B	$0.01 B
Total hospitalizations costs	−$13.08 B	−$13.10 B	$0.01 B
Total cost (US$)	−$14.24 B	−$11.38 B	−$2.85 B
Total QALYs	1,244,883	1,243,068	1815
ICER ($/QALY)	Dominant	Dominant	Dominant
Societal perspective			
Direct cost	−$1.70 B	$0.60 B	−$2.30 B
Indirect costs due to loss of productivity	−$205.59 B	−$205.40 B	−$0.19 B
Total cost (US$)	−$207.28 B	−$204.79 B	−$2.49 B
Total QALYs	1,244,883	1,243,068	1815
ICER ($/QALY)	Dominant	Dominant	Dominant

**Abbreviations:** B, billion; ED emergency department; ICER; incremental cost-effectiveness ratio; OP, outpatient; QALY, quality-adjusted life year.

## Data Availability

All data generated or analyzed during this study, which support the findings of this study, are included within this article and its Appendix A. Any data not present in the manuscript will be available from the corresponding author upon reasonable request.

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
