# Peer review of "A Clinical and Economic Comparison of Cell-Based Versus Recombinant Influenza Vaccines in Adults 18–64 Years in the United States"

_vaccines, 2024, doi:10.3390/vaccines12111217_

Round 1

Reviewer 1 Report

Comments and Suggestions for Authors

1)Why the model was include the children (0 to 17 years old)?Since the vaccine is for 18+

2)the commercial perspective, the authros means a payer perspective?

3)Table 1 is too big, please seperate the table at leat 3 with clinical epic data, cost data and utiliy

4)Vaccination coverage was the some for both vaccines?

5) what the resources include in  the Outpatient visit?

6)The cost of vaccine is per dose?

7)In table 2 the authors must reported that the table showes the difference!

Author Response

Thank you very much for taking the time to review our manuscript. Please find the detailed responses below and the corresponding revisions/corrections highlighted/in track changes in the re-submitted file (text added is marked with red font).

Comment 1: Why the model was include the children (0 to 17 years old)? Since the vaccine is for 18+

Response 1: Thanks for your comments. Our cost-effectiveness model focused on the clinical and economic outcomes among an 18–64-year-old population. We included all populations age groups in the compartmental model (SEIR) to estimate the number of infection cases in the US, considering herd immunity. The SEIR model accounts for the contact between the different age groups in the US population. Adults that are the main vaccination focus can infect children and older adult and also, they can get the infection from children or the elderly. Details about the SEIR objective and structure are listed in Methods (Line 99-123). Note: the vaccine is approved for children 6+ months, and in the SEIR model, we accounted for the rate of vaccination and acquired immunity among vaccinated children as well.

Comment 2: the commercial perspective, the authros means a payer perspective?

Response 2: Yes. The model assessed the outcomes from payer perspective (the commercial insurance companies) and the societal perspective as well. The statement is mentioned in line 95 of the manuscript.

Comment 3: Table 1 is too big, please seperate the table at leat 3 with clinical epic data, cost data and utiliy. 

Response 3: Thanks for the comment. We split the input table into two tables. The first for population and clinical inputs and the second for the costs and utilities inputs. 

Comment 4: Vaccination coverage was the some for both vaccines? 

Response 4: Yes, the overall vaccination coverage as sourced from the CDC in the US varies across the different age groups, but the model uses the same coverage rate for both vaccination arms to estimate the impact of each vaccine's effectiveness on reducing the total number of infections. We edited the sentence in line 199 of the manuscript: “US vaccination coverage rate data for the 2022-2023 season from the CDC [29] was used to estimate the number of vaccinated individuals in each age group, either vaccinated with a cell-based or recombinant vaccine.

Comment 5: what the resources include in the Outpatient visit? 

Response 5: Thanks for the comment. We didn’t estimate the cost by using a micro-costing approach of what resources are used in an OP visit, we used the physician’s services cost paid by commercial insurance of $107.10 for OP visit using the reference [the 2022 Congressional Budget Office report (table of inputs)]. For physicians’ services overall, commercial insurers paid 129% of Medicare OP physician service. [calculated as 129%* (OP visit cost in Medicare CPT 99203 that equals $83.02 = 107.10]. We added more details in the table of inputs to make it clearer.

Outpatient visit cost $107.10 (calculated as: CMS OP visit cost of $83.02 x commercial insurer pays 129% of Medicare OP physician fee service) $85.70, $128.50 Congressional Budget Office 2022[39], CMS physician fee schedule CPT 99203[40]  

Comment 6:The cost of vaccine is per dose? 

Response 6: Yes. WAC per dose was added to the inputs table. [Reference for WAC: RedBook Micromedex 2023]

Comment 7: In table 2 the authors must reported that the table showes the difference!

Response 7: Table 2 (now table 3) shows differences in costs and QALYs between cell-based and no vaccine; recombinant and no vaccine; and cell-based vs. recombinant.  But not all results are differences, the ICER is not a difference, so we suggest to keep the title of the table as Cost-effectiveness results season 2018-2019.  In section 3.1 of the text, it is reported that values are differences where appropriate.

Reviewer 2 Report

Comments and Suggestions for Authors

Manuscript ID: Vaccines-3232885

Title: A Clinical and Economic Comparison of Cell-based Versus Recombinant Influenza Vaccine in Adults 18-64 Years in the United States.

COMMENTS TO THE AUTHORS

General comments

The authors evaluated the cost-effectiveness of a cell-based influenza vaccine to a recombinant influenza vaccine, and each to no vaccination using US commercial and societal perspectives. This is a well-done and important study. I only have a few comments/suggestions below.

Specific comments for revision:

a)      Major

    • Line 129: it’s not clear how the authors categorized groups with high vs. low risk of developing influenza-related complications. This is important. Please add a sentence for how they are defined and/or a reference.
    • Table2: I suggest that the authors report $$$ in same unit in the second and third columns of table 2 (especially the costs from the societal perspective). This way, it’ll be easy and quick for the readers to compare and also to be consistent with the text in lines 279-280.

b)      Minor

    • Line 48: typo.
    • Delete extra space between words throughout the manuscript (example Lines 58, 79, 88).
    • Please insert space in 95%CI throughout the manuscript.
    • Line 244: results “are” reported.
    • Table 2: Please use 1- or 2-digits only after decimal point.
    • Please be consistent in using use 1- or 2-digits after decimal point throughout the text.

Author Response

Thank you very much for taking the time to review our manuscript. Please find the detailed responses below and the corresponding revisions/corrections highlighted/in track changes in the re-submitted file (added text is marked with red font)

Comment 1: The authors evaluated the cost-effectiveness of a cell-based influenza vaccine to a recombinant influenza vaccine, and each to no vaccination using US commercial and societal perspectives. This is a well-done and important study. I only have a few comments/suggestions below. 

Response 1: Thank you for your positive feedback

Comment 2: Line 129: it’s not clear how the authors categorized groups with high vs. low risk of developing influenza-related complications. This is important. Please add a sentence for how they are defined and/or a reference.

Response 2: Thanks for the valuable feedback. The definitions are added in line 133. “Cases were categorized based on the risk (high or low) of developing influenza-related complications (Figure 1). High risk was identified by the presence of one or more chronic conditions, including cardiovascular, respiratory, renal, or immune-related diseases. Low risk was identified if patients did not have any of the high-risk chronic conditions. [19]” [Reference 19: Zimmerman RK, Dauer K, Clarke L, et al. Vaccine effectiveness of recombinant and standard dose influenza vaccines against outpatient illness during 2018-2019 and 2019-2020 calculated using a retrospective test-negative design. Hum Vaccin Immunother. 2023 Dec 31;19(1):2177461. doi: 10.1080/21645515.2023.2177461.

Comment 3: Table2: I suggest that the authors report $$$ in same unit in the second and third columns of table 2 (especially the costs from the societal perspective). This way, it’ll be easy and quick for the readers to compare and also to be consistent with the text in lines 279-280.

Response 3: Thanks for the comment. Yes, it makes more sense to report all costs results using same unit. We made the required edit, and all costs are reported in billion unit. 

Minor comments:   Thank you for catching these items, please see below our edits:

Comment 4: Line 48: typo.

Response 4: Corrected: “The annual economic burden of influenza on the US healthcare system has been estimated”

Comment 5: Delete extra space between words throughout the manuscript (example Lines 58, 79, 88). 

Response 5: All extra spaces have been removed throughout the manuscript.

Comment 6: Please insert space in 95%CI throughout the manuscript.

Response 6: Spaces have been added throughout the manuscript. 

Comment 7: Line 244: results “are” reported.

Response 7: Changed from 'were' to 'are'

Comment 8: Table 2: Please use 1- or 2-digits only after decimal point.

Response 8: Changed to two decimal point format.

Comment 9: Please be consistent in using use 1- or 2-digits after decimal point throughout the text.

Response 9: Required edits have been made; 2 decimal point format has been used.

Reviewer 3 Report

Comments and Suggestions for Authors

This paper reports an analysis of the cost-effectiveness of a cell-based influenza vaccine to a recombinant influenza vaccine, and each to no vaccination for data for 18-64 year-olds in the United States from the 2018-2019 influenza season. Here are some things to attend to in a revision.

First, the specific influenza season studied in the paper needs to be added to the title.

Second, the Abstract, Introduction, and Discussion sections of the paper need to more clearly state the focus of the paper (as noted above) -- and to clearly and specifically state the particular contributions of the research described in the paper to the existing prior research publications. Specifically, state what is unique and significant in this paper, namely, the focus on cost-effectiveness.

Third, the acronym ICERs is used in the Abstract and is not clearly stated verbally until lines 137-138. Therefore, it needs to be spelled out in the Abstract. Indeed, the paper uses many acronyms, so it would be useful to readers for their reference to include a table wherein all are presented.

Author Response

Thank you very much for taking the time to review our manuscript. Please find the detailed responses below and the corresponding revisions/corrections highlighted/in track changes in the re-submitted file (added text is marked with red font)

Comment 1: First, the specific influenza season studied in the paper needs to be added to the title. 

Response 1: Thanks for your suggestion. We prefer to keep the current title as the study includes one additional season (2019-2020) in the scenario analysis, and we want to avoid sending a message of results being relevant only for one single season.

Comment 2: Second, the Abstract, Introduction, and Discussion sections of the paper need to more clearly state the focus of the paper (as noted above) -- and to clearly and specifically state the particular contributions of the research described in the paper to the existing prior research publications. Specifically, state what is unique and significant in this paper, namely, the focus on cost-effectiveness.

Response 2: Thanks for the suggestion, we did some edits on the manuscript as described below:

1) We added more details in the introduction: “Understanding the cost-effectiveness profile of non-egg-based influenza vaccines is essential to guide payers in allocating resources efficiently and prioritizing vaccines that offer the best health outcomes for the funds available. Due to lack of published evidence we conducted this study to evaluate the cost-effectiveness of the cell-based influenza vaccine compared to the recombinant influenza vaccine - focusing on the prevention of influenza-related outpatients visits, emergency department (ED) visits, and hospitalizations among US adults aged 18 – 64 years - from US commercial payer and societal perspectives using absolute and relative vaccines effectiveness for season 2018-2019.”

2) we added more text in the discussion: “Non-egg-based influenza vaccines have been approved for vaccinating children, adults, and the elderly. Several real-world studies have shown improved clinical outcomes with these vaccines compared to egg-based vaccines. However, no evidence has compared the effectiveness and economic consequences between the two non-egg-based vaccines (cell-based vs. recombinant vaccine). Therefore, we aimed to conduct cost-effectiveness analyses from the US payer and societal perspectives in adults aged 18 to 64, with or without high risk of developing complications if infected, using vaccine effectiveness data from the 2018-2019 season as the base case.”

Comment 3: Third, the acronym ICERs is used in the Abstract and is not clearly stated verbally until lines 137-138. Therefore, it needs to be spelled out in the Abstract. Indeed, the paper uses many acronyms, so it would be useful to readers for their reference to include a table wherein all are presented. 

Response 3: Now the acronym is spelled out in the abstract. Thanks so much for your suggestion, this is highly value.

Round 2

Reviewer 3 Report

Comments and Suggestions for Authors

The revisions to this manuscript have been responsive to the previous review, and it has been improved accordingly. I have no further suggestions for revision.